# Multi-scale Graphical Models for Spatio-Temporal Processes

**Firdaus Janoos**[*]      **Huseyin Denli**      **Niranjan Subrahmanya**
ExxonMobil Corporate Strategic Research
Annandale, NJ 08801

## Abstract

Learning the dependency structure between spatially distributed observations of a spatio-temporal process is an important problem in many fields such as geology, geophysics, atmospheric sciences, oceanography, *etc.* . However, estimation of such systems is complicated by the fact that they exhibit dynamics at multiple scales of space and time arising due to a combination of diffusion and convection/advection [17]. As we show, time-series graphical models based on vector auto-regressive processes[18] are inefficient in capturing such multi-scale structure. In this paper, we present a hierarchical graphical model with physically derived priors that better represents the multi-scale character of these dynamical systems. We also propose algorithms to efficiently estimate the interaction structure from data. We demonstrate results on a general class of problems arising in exploration geophysics by discovering graphical structure that is physically meaningful and provide evidence of its advantages over alternative approaches.

## 1   Introduction

Consider the problem of determining the connectivity structure of subsurface aquifers in a large ground-water system from time-series measurements of the concentration of tracers injected and measured at multiple spatial locations. This problem has the following features: **(i)** pressure gradients driving ground-water flow have unmeasured disturbances and changes; **(ii)** the data contains only concentration of the tracer, not flow direction or velocity; **(iii)** there are regions of high permeability where ground water flows at (relatively) high speeds and tracer concentration is conserved and transported over large distances **(iv)** there are regions of low permeability where ground water diffuses slowly into the bed-rock and the tracer is dispersed over small spatial scales and longer time-scales.

Reconstructing the underlying network structure from spatio-temporal data occurring at multiple spatial and temporal scales arises in a large number of fields. An especially important set of applications arise in exploration geophysics, hydrology, petroleum engineering and mining where the aim is to determine the connectivity of a particular geological structure from sparsely distributed time-series readings [16]. Examples include exploration of ground-water systems and petroleum reservoirs from tracer concentrations at key locations, or use of electrical, induced-polarization and electro-magnetic surveys to determine networks of ore deposits, groundwater, petroleum, pollutants and other buried structures [24]. Other examples of multi-scale spatio-temporal phenomena with the network structure include: flow of information through neural/brain networks [15], traffic flow through traffic networks[3]; spread of memes through social networks [23]; diffusion of salinity, temperature, pressure and pollutants in atmospheric sciences and oceanography [9]; transmission networks for genes, populations and diseases in ecology and epidemiology; spread of tracers and drugs through biological networks [17] *etc.* .

---

[*]Corresponding Author:firdaus@ieee.org

These systems typically exhibit the following features: **(i)** the physics are linear in the observed / state variables (*e.g.* pressure, temperature, concentration, current) but non-linear in the unknown parameter that determines interactions (*e.g.* permeability, permittivity, conductance); **(ii)** there may be unobserved / unknown disturbances to the system; **(iv)** *(Multi-scale structure)* there are interactions occurring over large spatial scales versus those primarily in local neighborhoods. Moreover, the large-scale and small-scale processes exhibit characteristic time-scales determined by the balance of convection velocity and diffusivity of the system. A physics-based approach to estimating the structure of such systems from observed data is by inverting the governing equations [1]. However, in most cases inversion is extremely ill-posed [21] due to non-linearity in model parameters and sparsity of data with respect to the size of the parameter space, necessitating strong priors on the solution which are rarely available. In contrast, there is a large body of literature on structure learning for time-series using data-driven methods, primarily developed for econometric and neuroscientific data[1]. The most common approach is to learn vector auto-regressive (VAR) models, either directly in the time domain[10] or in the frequency domain[4]. These implicitly assume that all dynamics and interactions occur at similar time-scales and are acquired at the same frequency [14], although VAR models for data at different sampling rates have also been proposed [2]. These models, however, do not address the problem of interactions occurring at multiple scales of space and time, and as we show, can be very inefficient for such systems. Multi-scale graphical models have been constructed as pyramids of latent variables, where higher levels aggregate interactions at progressively larger scales [25]. These techniques are designed for regular grids such as images, and are not directly applicable to unstructured grids, where spatial distance is not necessarily related to the dependence between variables. Also, they construct $\mathcal{O}(\log N)$ deep trees thereby requiring an extremely large ($\mathcal{O}(N)$) latent variable space.

In this paper, we propose a new approach to learning the graphical structure of a multi-scale spatio-temporal system using a hierarchy of VAR models with one VAR system representing the large-scale (global) system and one VAR-X model for the (small-scale) local interactions. The main contribution of this paper is to model the global system as a flow network in which the observed variable both convects and diffuses between sites. *Convection-diffusion* (C–D) processes naturally exhibit multi-scale dynamics [8] and although at small spatial scales their dynamics are varied and transient, at larger spatial scales these processes are smooth, stable and easy to approximate with coarse models [13]. Based on this property, we derive a regularization that replicates the large-scale dynamics of C–D processes. The hierarchial model along with this physically derived prior learns graphical structures that are not only extremely sparse and rich in their description of the data, but also physically meaningful. The multi-scale model both reduces the number of edges in the graph by clustering nodes and also has smaller order than an equivalent VAR model. Next in Section 3, model relaxations to simplify estimation along with efficient algorithms are developed. In Section 4, we present an application to learning the connectivity structure for a class of problems dealing with flow through a medium under a potential/pressure field and provide theoretical and empirical evidence of its advantages over alternative approaches.

One similar approach is that of clustering variables while learning the VAR structure [12] using sampling-based inference. This method does not, however, model dynamical interactions between the clusters themselves. Alternative techniques such as independent process analysis [20] and AR-PCA [7] have also been proposed where auto-regressive models are applied to latent variables obtained by ICA or PCA of the original variables. Again, because these are AR not **V**AR models, the interactions between the latent variables are not captured, and moreover, they do not model the dynamics of the original space. In contrast to these methods, the main aspects of our paper are a hierarchy of dynamical models where each level explicitly corresponds to a spatio-temporal scale along with efficient algorithms to estimate their parameters. Moreover, as we show in Section 4, the prior derived from the physics of C–D processes is critical to estimating meaningful multi-scale graphical structures.

## 2   Multi-scale Graphical Model

**Notation:** Throughout the paper, upper case letters indicate matrices and lower-case boldface for vectors, subscript for vector components and $[t]$ for time-indexing.

Let $\mathbf{y} \in \mathbb{R}^{N \times T}$, where $\mathbf{y}[t] = \{\mathbf{y}_1[t] \ldots \mathbf{y}_N[t]\}$; $t = 1 \ldots T$, be the time-series data observed at $N$ sites over $T$ time-points. To capture the multi-scale structure of interactions at local and global scales, we introduce the $K$–dimensional ($K \ll N$) latent process $\mathbf{x}[t] = \{\mathbf{x}_1[t] \ldots \mathbf{x}_K[t]\}$; $t = 1 \ldots T$ to represent $K$ global components that interact with each other. Each observed process $\mathbf{y}_i$ is then a summation of local interactions along with a global interaction. Specifically:

$$
\begin{aligned}
\textbf{Global–process:} \quad & \mathbf{x}[t] = \sum_{p=1}^{P} \mathrm{A}[p]\mathbf{x}[t-p] + \mathbf{u}[t], \\
\textbf{Local–process:} \quad & \mathbf{y}[t] = \sum_{q=1}^{Q} \mathrm{B}[q]\mathbf{y}[t-q] + \mathrm{Z}\mathbf{x}[t] + \mathbf{v}[t].
\end{aligned}
\tag{1}
$$

Here $\mathrm{Z}_{i,k}$, $i = 1 \ldots N$, $k = 1 \ldots K$ are binary variables indicating if site $\mathbf{y}_i$ belongs to global component $\mathbf{x}_k$. The $N \times N$ matrices $\mathrm{B}[1] \ldots \mathrm{B}[Q]$ capture the graphical structure and dynamics of the local interactions between all $\mathbf{y}_i$ and $\mathbf{y}_j$, while the set of $K \times K$ matrices $\mathrm{A} = \{\mathrm{A}_1 \ldots \mathrm{A}[P]\}$ determines the large-scale graphical structure as well as the overall dynamical behavior of the system. The processes $\mathbf{v} \sim \mathcal{N}(0, \sigma_{\mathbf{v}}^2 \mathrm{I})$ and $\mathbf{u} \sim \mathcal{N}(0, \sigma_{\mathbf{u}}^2 \mathrm{I})$ are iid innovations injected into the system at the global and local scale respectively.

**Remark:** From a graphical perspective, two latent components $\mathbf{x}_k$ and $\mathbf{x}_l$ are conditionally independent given all other components $\mathbf{x}_m$, $\forall m \neq k, l$ if and only if $\mathrm{A}[p]_{i,j} = 0$ for all $p = 1 \ldots P$. Moreover, two nodes $\mathbf{y}_i$ and $\mathbf{y}_j$ are conditionally independent given all other nodes $\mathbf{y}_m \neq i, j$ and latent components $\mathbf{x}_k, \forall k = 1 \ldots K$, if and only if $\mathrm{B}[q]_{i,j} = 0$ for all $q = 1 \ldots Q$.

To create the multi-scale hierarchy in the graphical structure, the following two conditions are imposed: **(i)** each $\mathbf{y}_i$ belong to only one global component $\mathbf{x}_k$, *i.e.* $\mathrm{Z}_{i,k}\mathrm{Z}_{i,l} = \delta[k,l]$, $\forall i = 1 \ldots N$; and **(ii)** $\mathrm{B}_{i,j}$ be non-zero only for nodes within the same component, *i.e.* $\mathrm{B}_{i,j} = 0$ if $\mathbf{y}_i$ and $\mathbf{y}_j$ belong to different global components $\mathbf{x}_k$ and $\mathbf{x}_{k'}$.

The advantages of this model over a VAR graphical model are two fold: **(i)** the hierarchical structure, the fact that $K \ll N$ and that $\mathbf{y}_i \leftrightarrow \mathbf{y}_j$ only if they are in the same global component results in a very sparse graphical model with a rich multi-scale interpretation; and **(ii)** as per Theorem 1, the model of eqn. (1) is significantly more parsimonious than an equivalent VAR model for data that is inherently multi-scale.

**Theorem 1.** *The model of eqn. (1) is equivalent to a vector auto-regressive moving-average (VARMA) process* $\mathbf{y}[t] = \sum_{r=1}^{R} \mathrm{D}[r]\mathbf{y}[t-r] + \sum_{s=0}^{S} \mathrm{E}[s]\epsilon[t-s]$ *where* $P \leq R \leq P + Q$ *and* $0 \leq S \leq P$, $\mathrm{D}[r]$ *are* $N \times N$ *full-rank matrices and* $\mathrm{E}[s]$ *are* $N \times N$ *matrices with rank less than* $K$. *Moreover the upper bounds are tight if the model of eqn. (1) is minimal. The proof is given in Supplemental Appendix A.*

The multi-scale spatio-temporal dynamics are modeled as stable *convection–diffusion* (C–D) processes governed by *hyperbolic–parabolic* PDEs of the form $\partial y / \partial t + \nabla \cdot (\vec{\mathbf{c}} y) = \nabla \cdot \kappa \nabla + s$, where $y$ is the quantity corresponding to $\mathbf{y}$, $\kappa$ is the diffusivity and $\mathbf{c}$ is the convection velocity and $s$ is an exogenous source. The balance between convection and diffusion is quantified by the Péclet number[2] of the system [8]. These processes are non-linear in diffusivity and velocity and a full-physics inversion involves estimating $\kappa$ and $\vec{\mathbf{c}}$ at each spatial location, which is a highly ill-posed and under-constrained[1]. However, because for systems with physically reasonable Péclet numbers, dynamics at larger scales can be accurately approximated on increasingly coarse grids [13], we simplify the model by assuming that conditioned on the rest of the system, the large-scale dynamics between any two components $\mathbf{x}_i \sim \mathbf{x}_j \mid \mathbf{x}_k \; \forall k \neq i, j$ can be approximated by a 1-d C–D system with constant Péclet number. This approximation allows us to use Proposition 2:

**Theorem 2.** *For the VAR system of eqn. (1), if the dynamics between any two variables* $\mathbf{x}_i \sim \mathbf{x}_j \mid \mathbf{x}_k \; \forall k \neq i, j$ *are 1–d C–D with infinite boundary conditions and constant Péclet number, then the VAR coefficients* $\mathrm{A}_{i,j}[t]$ *can be approximated by a Gaussian function* $\mathrm{A}_{i,j}[t] \approx \exp\left\{-0.5(t - \mu_{i,j})^2 \sigma_{i,j}^{-2}\right\} / \sqrt{2\pi\sigma_{i,j}^2}$ *where* $\mu_{i,j}$ *is equal to the distance between* $i$ *and* $j$ *and* $\sigma_{i,j}^2$ *is proportional to the product of the distance and the Péclet number. Moreover, this approximation has a multiplicative-error* $\exp(-\mathcal{O}(t^3))$. *Proof is given in Supplemental Appendix B.*

In effect, the dynamics of a multi-dimensional (*i.e.* 2-d or 3-d) continuous spatial system are approximated as a network of 1-dimensional point-to-point flows consisting of a combination of advection

and diffusion. Although in general, the dynamics of higher-dimensional physical systems are not equivalent to super-position of lower-dimensional systems, as we show in this paper, the stability of C–D physics [13] allows replicating the large-scale graphical structure and dynamics, while avoiding the ill-conditioned and computationally expensive inversion of a full-physics model. Moreover, the stability of the C–D impulse response function ensures that the resulting VAR system is also stable.

## 3  Model Relaxation and Regularization

As the model of eqn. (1) contains non-linear interactions of real-valued variables $\mathbf{x}$, A and B with binary Z along with mixed constraints, direct estimation would require solving a mixed integer non-linear problem. Instead, in this section we present relaxations and regularizations that allow estimation of model parameters via convex optimization. The next theorem states that for a given assignment of measurement sites to global components, the interactions within a component do not affect the interactions between components, which enables replacing the mixed non-linearity due to the constraints on $\mathrm{B}[q]$ with a set of unconstrained diagonal matrices $\mathrm{C}[q]$, $q = 1 \ldots Q$.

**Theorem 3.** *For a given global-component assignment* Z*, if* $\mathrm{A}^*$ *and* $\mathbf{x}^*$ *are local optima to the least-squares problem of eqn.* (1)*, then they are also a local optimum to the least-squares problem for:*

$$\mathbf{x}[t] = \sum_{p=1}^{P} \mathrm{A}[p]\mathbf{x}[t-p] + \mathbf{u}[t] \qquad and \qquad \mathbf{y}[t] = \sum_{q=1}^{Q} \mathrm{C}[p]\mathbf{y}[t-q] + \mathrm{Z}\mathbf{x}[t] + \mathbf{v}[t], \tag{2}$$

*where* $\mathrm{C}[r]$, $r = 1 \ldots b$ *are diagonal matrices. The proof is given in Supplemental Appendix C.*

Furthermore, a LASSO regularization term proportional $\|\mathrm{C}\|_1 = \sum_{i=1}^{N} \sum_{q=1}^{Q} |\mathrm{C}[q][i,i]|$ is added to reduce the number of non-zero coefficients and thereby the effective order of C .

Next, the binary indicator variables $\mathrm{Z}_{i,k}$ are relaxed to be real-valued. Also, an $\ell_1$ penalty, which promotes sparsity, combined with an $\ell_2$ term has been shown to estimate disjoint clusters[19]. Therefore, the spatial disjointedness constraint $\mathrm{Z}_{i,k}\mathrm{Z}_{i,l} = \delta_{k,l}$, $\forall i = 1 \ldots N$, is relaxed by a penalty proportional to $\|\mathrm{Z}_{i,\cdot}\|_1$ along with the constraint that for each $\mathbf{y}_i$, the indicator vector $\mathrm{Z}_{i,\cdot}$ should lie within the unit sphere, *i.e.* $\|\mathrm{Z}_{i,\cdot}\|_2 \leq 1$. This penalty, which also ensures that $|\mathrm{Z}_{i,k}| \leq 1$, allows interpretation of $\mathrm{Z}_{i,\cdot}$ as a soft cluster membership.

One way to regularize $\mathrm{A}_{i,j}$ according to Theorem 2 would be to directly parameterize it as a Gaussian function. Instead, observe that $G(t) = \exp\left\{-0.5(t-\mu)^2/\sigma^2\right\}/\sqrt{2\pi\sigma^2}$ satisfies the equation $[\partial_t + (t-\mu)/\sigma]\,G = 0$, subject to $\int G(t)dt = 1$. Therefore, defining the discrete version of this operator as $\mathrm{D}(\gamma_{i,j})$, a $P \times P$ diagonal matrix, the regularization A is as a penalty proportional to

$$\|\mathrm{D}(\gamma)\mathrm{A}\|_{2,1} = \sum_{i,j} \|\mathrm{D}(\gamma_{i,j})\mathrm{A}_{i,j}\|_2 \quad \text{where} \quad \mathrm{D}\left(\gamma_{i,j}\right)_{p,p} = \widehat{\partial}_p + \gamma_{i,j}\left(p - \mu_{i,j}\right), \tag{3}$$

along with the relaxed constraint $0 \leq \sum_p \mathrm{A}_{i,j}[p] \leq 1$. Here, $\widehat{\partial}_p$ is an approximation to time-differentiation, $\mu_{i,j}$ is equal to the distance between $i$ and $j$ *which is known*, and $\gamma_{i,j} \geq \Gamma$ is inversely proportional to $\sigma_{i,j}$. Importantly, this formulation also admits 0 as a valid solution and has two advantages over direct parametrization: **(i)** it replaces a problem that is non-linear in $\sigma_{i,j}^2$ ; $i, j = 1 \ldots K$ with a penalty that is linear in $\mathrm{A}_{i,j}$; and **(ii)** unlike Gaussian parametrization, it admits the sparse solution $\mathrm{A}_{i,j} = 0$ for the case when $\mathbf{x}_i$ does not directly affect $\mathbf{x}_j$. The constant $\Gamma > 0$ is a user-specified parameter which prevents $\gamma_{i,j}$ from taking on very small values, thereby obviation solutions of $\mathrm{A}_{i,j}$ with extremely large variance *i.e.* with very small but non-zero value. This penalty, derived from considerations of the dynamics of multi-scale spatio-temporal systems, is the key difference of the proposed method as compared to sparse time-series graphical model via group LASSO [11].

Putting it all together, the multi-scale graphical model is obtained by optimizing:

$$[\mathbf{x}^*, \mathrm{A}^*, \mathrm{C}^*, \mathrm{Z}^*, \gamma^*] = \underset{\mathbf{x}, \mathrm{A}, \mathrm{C}, \mathrm{Z}, \gamma}{\operatorname{argmin}} \; f(\mathbf{x}, \mathrm{A}, \mathrm{C}, \mathrm{Z}, \gamma) + g(\mathbf{x}, \mathrm{A}, \mathrm{C}, \mathrm{Z}) \tag{4}$$

subject to $\|\mathrm{Z}_{i,\cdot}\|_2^2 \leq 1$ for all $i = 1 \ldots N$ and $0 \leq \sum_p \mathrm{A}_{i,j}[p] \leq 1$ for all $i, j = 1 \ldots K$, and $\gamma_{i,j} \geq \Gamma$, $\forall i, j = 1 \ldots K$. The objective function is split into a smooth portion :

$$f(\mathbf{x}, \theta) = \sum_{t=1}^{T} \left\| \mathbf{y}[t] - \sum_{q=1}^{Q} \mathrm{C}[q]\mathbf{y}[t-q] - \mathrm{Z}\mathbf{x}[t] \right\|_2^2 + \lambda_0 \left\| \mathbf{x}[t] - \sum_{p=1}^{P} \mathrm{A}[p]\mathbf{x}[t-p] \right\|_2^2$$

and a non-smooth portion $g(\theta) = \lambda_1 \|\mathrm{D}(\gamma)\mathrm{A}\|_{2,1} + \lambda_2 \|\mathrm{C}\|_{2,1} + \lambda_3 \|\mathrm{Z}\|_1$. After solving eqn. (4), the local graphical structure within each global component is obtained by solving: $\mathrm{B}^* = \mathrm{argmin}_{\mathrm{B}} \sum_{t=1}^{T} \left\| \mathbf{y}[t] - \sum_{q=1}^{Q} \mathrm{B}[q]\mathbf{y}[t-q] - \mathrm{Z}^*\mathbf{x}^*[t] \right\|_2^2 + \lambda_4 \|\mathrm{B}\|_{2,1}$, where the zeros of $\mathrm{B}[q]$ are pre-determined from $\mathrm{Z}^*$.

## 3.1 Optimization

Given values of $[\mathrm{A}, \mathrm{Z}, \mathrm{C}]$, the problem of eqn. (4) is unconstrained and strictly convex in $\mathbf{x}$ and $\gamma$ and given $[\mathbf{x}, \gamma]$, it is unconstrained and strictly convex in C and convex constrained in A and Z. Therefore, under these conditions *block coordinate descent* (BCD) is guaranteed to produce a sequence of solutions that converge to a stationary point [22]. To avoid saddle-points and achieve local-minima, a random feasible-direction heuristic is used at stationary points. Defining blocks of variables to be $[\mathbf{x}, \gamma]$, and $[\mathrm{A}, \mathrm{C}, \mathrm{Z}]$, BCD operates as follows:

1 Initialize $\mathbf{x}^{(0)}$ and $\gamma^{(0)}$

2 Set $n = 0$ and repeat until convergence:

$$[\mathrm{A}^{(n+1)}, \mathrm{Z}^{(n+1)}, \mathrm{C}^{(n+1)}] \leftarrow \min_{[\mathrm{A}, \mathrm{Z}, \mathrm{C}]} f(\mathbf{x}^{(n)}, \mathrm{A}, \mathrm{C}, \mathrm{Z}, \gamma^{(n)}) + g(\mathbf{x}^{(n)}, \mathrm{A}, \mathrm{C}, \mathrm{Z})$$

$$[\mathbf{x}^{(n+1)}, \gamma^{(n+1)}] \leftarrow \min_{[\mathbf{x}, \gamma]} f(\mathbf{x}, \mathrm{A}^{(n+1)}, \mathrm{C}^{(n+1)}, \mathrm{Z}^{(n+1)}, \gamma) + g(\mathbf{x}, \mathrm{A}^{(n+1)}, \mathrm{C}^{(n+1)}, \mathrm{Z}^{(n+1)}).$$

At each iteration $\mathbf{x}^{(n+1)}$ is obtained by directly solving a $T \times T$ tri-diagonal Toeplitz system with blocks of size $KP$ which has a have running time of $\mathcal{O}(T \times KP^3)$ (§Supplemental Appendix D for details).

Estimating $\gamma^{(n+1)}$ given $\mathrm{A}^{(n+1)}$ is obtained by solving $\min_{\gamma_{i,j}} \sum_{p=1}^{P} \left( \widehat{\partial}_p \mathrm{A}_{i,j}[p] + \gamma_{i,j} \left( p - \mu_{i,j} \right) \mathrm{A}_{i,j}[p] \right)^2$ subject to $\gamma_{i,j} > \Gamma$ for all $i, j = 1 \ldots K$ and $i \neq j$. This gives $\gamma_{i,j}^{(n+1)} = \max \left( \Gamma, \ -\sum_p \partial_t \mathrm{A}_{i,j} \left( p - \mu_{i,j} \right) \mathrm{A}_{i,j} / \sum_p ((p - \mu_{i,j}) \, \mathrm{A}_{i,j})^2 \right)$.

Optimization with respect to $\mathrm{A}, \mathrm{Z}, \mathrm{C}$ is performed using proximal splitting with Nesterov acceleration [5] which produces $\epsilon$–optimal solutions in $\mathcal{O}(1/\sqrt{\epsilon})$ time, where the constant factor depends on $\sqrt{L(\nabla_\theta f)}$, the Lipschitz constant of the gradient of the smooth portion $f$. Defining $\theta = [\mathrm{A}, \mathrm{Z}, \mathrm{C}]$, the key step in the optimization are proximal-gradient-descent operations of the form: $\theta^{(m)} = \mathbf{prox}_{\alpha_m g} \left( \theta^{(m-1)} - \alpha_m \nabla_\theta f \left( \mathbf{x}^{(n)}, \gamma^{(n)}, \theta^{(m-1)} \right) \right)$, where $m$ is the current gradient-descent iterate, $\alpha_m$ is the step size and the proximal operator is defined as: $\mathbf{prox}_g(\theta) = \min_\theta g(\mathbf{x}^{(n)}, \gamma^{(n)}, \theta) + \frac{1}{2} \|\theta - \Theta\|^2$.

The gradients $\nabla_\mathrm{A} f$, $\nabla_\mathrm{C} f$ and $\nabla_\mathrm{Z} f$ are straightforward to compute. As shown in Supplemental Appendix E.1, the problem in Z is decomposable into a sum of problems over $\mathrm{Z}_{i,\cdot}$ for $i = 1 \ldots N$, where the proximal operator for each $\mathrm{Z}_{i,\cdot}$ is $\mathbf{prox}_g(\mathrm{Z}_{i,\cdot}) = \max\left(1, \|T_\lambda(\mathrm{Z}_{i,\cdot})\|_2^{-1}\right) T_\lambda(\mathrm{Z}_{i,\cdot})$. Here $T_{\lambda_3}(\mathrm{Z}_{i,k}) = \mathrm{sign}(\mathrm{Z}_{i,k}) \min(|\mathrm{Z}_{i,k}| - \lambda_3, 0)$ is the element-wise shrinkage operator.

Because A has linear constraints of the form $0 \leq \sum_p \mathrm{A}_{i,j}[p] \leq 1$, the proximal operator does not have a closed form solution and is instead computed using dual-ascent [6]. As it can be decomposed across $\mathrm{A}_{i,j}$ for all $i, j = 1 \ldots K$, consider the computation of $\mathbf{prox}_g(\hat{\mathbf{a}})$ where $\hat{\mathbf{a}}$ represents one $\mathrm{A}_{i,j}$. Defining $\eta$ as the dual variable, dual-ascent proceeds by iterating the following two steps until convergence:

$$\textbf{(i):} \quad \mathbf{a}^{(n+1)} = \begin{cases} \hat{\mathbf{a}} + \eta^{(n)}\mathbb{1} - \lambda \frac{\hat{\mathbf{a}} + \eta^{(n)}\mathbb{1}}{\|\mathrm{D}^{-1}\hat{\mathbf{a}} + \eta^{(n)}\mathbb{1}\|_2} & \text{if} \quad \left\|\mathrm{D}^{-1}\hat{\mathbf{a}} + \eta^{(n)}\mathbb{1}\right\|_2 > \lambda \\ 0 & \text{otherwise} \end{cases}$$

$$\textbf{(ii):} \quad \eta^{(n+1)} = \begin{cases} \eta^{(n)} - \alpha^{(n)}\mathbb{1}^\top \mathbf{a}^{(n+1)} & \text{if} \quad \mathbb{1}^\top \mathbf{a}^{(n+1)} < 0 \\ \eta^{(n)} + \alpha^{(n)}\left(\mathbb{1}^\top \mathbf{a}^{(n+1)} - 1\right) & \text{if} \quad \mathbb{1}^\top \mathbf{a}^{(n+1)} > 1 \end{cases}.$$

Here $n$ indexes the dual-ascent inner loop and $\alpha^{(n)}$ is an appropriately chosen step-size. Note that $\mathrm{D}(\gamma_{i,j})$, the $P \times P$ matrix approximation to $\partial_t + \gamma_{i,j}t$ is full rank and therefore invertible. And finally, the proximal operator for $\mathrm{C}_{i,i}$ for all $i = 1 \ldots N$ is $\mathrm{C}_{i,i} - \lambda_2 \mathrm{C}_{i,i}/\|\mathrm{C}_{i,i}\|_2$ if $\|\mathrm{C}_{i,i}\|_2 > \lambda_2$ and 0 otherwise.

**Remark:** The hyper-parameters of the systems are multipliers $\lambda_0 \ldots \lambda_4$ and threshold $\Gamma$. The term $\lambda_0$, which is proportional to $\sigma_{\mathbf{u}}/\sigma_{\mathbf{v}}$, implements a trade-off between innovations in the local and global processes. The parameter $\lambda_1$ penalizes deviation of $A_{i,j}$ from expected C–D dynamics, while $\lambda_2$, $\lambda_3$ and $\lambda_4$ control the sparsity of C, Z and B respectively. As explained earlier $\Gamma > 0$, the lower bound on $\gamma_{i,j}$, prohibits estimates of $A_{i,j}$ with very high variance and thereby controls the spread / support of A.

**Hyper-parameter selection:** Hyper-parameter values that minimize cross-validation error are obtained using grid-search. First, solutions over the full regularization path are computed with warm-starting. In our experience, for sufficiently small step sizes warm-starting leads to convergence in a few ($< 5$) iterations regardless of problem size. Moreover, as B is solved in a separate step, selection of $\lambda_4$ is done independently of $\lambda_0 \ldots \lambda_3$. Experimentally, we have observed that an upper limit on $\Gamma = 1$ and step-size of $0.1$ is sufficient to explore the space of all solutions. The upper limit on $\lambda_3$ is the smallest value for which any indicator vector $Z_{i,\cdot}$ becomes all zero. Guidance about minimum and maximum values $\lambda_0$ is obtained using the system identification technique of auto-correlation least squares.

**Initialization:** To cold start the BCD, $\gamma_{i,j}^{(0)}$ is initialized with the upper bound $\Gamma = 1$ for all $i, j = 1 \ldots K$. The variables $\mathbf{x}_1^{(0)} \ldots \mathbf{x}_K^{(0)}$ are initialized as centroids of clusters obtained by $K-$means on the time-series data $\mathbf{y}_1 \ldots \mathbf{y}_N$.

**Model order selection:** Because of the sparsity penalties, the solutions are relatively insensitive to model order $(P, Q)$. Therefore, these are typically set to high values and the effective model order is controlled through the sparsity hyper-parameters.

## 4   Results

In this section we present an application to determining the connectivity structure of a medium from data of flow through it under a potential/pressure field. Such problems include flow of fluids through porous media under pressure gradients, or transmission of electric currents through resistive media due to potential gradients, and commonly arise in exploration geophysics in the study of sub-surface systems like aquifers, petroleum reservoirs, ore deposits and geologic bodies [16]. Specifically, these processes are defined by PDEs of the form:

$$\vec{\mathbf{c}} + \kappa \nabla \cdot p = \mathbf{0} \qquad \text{and} \qquad \frac{\partial y}{\partial t} + \nabla (y\vec{\mathbf{c}}) = s_y, \tag{5}$$

$$\text{where} \qquad \nabla \cdot \vec{\mathbf{c}} = s_q \qquad \text{and} \qquad \vec{\mathbf{n}} \cdot \nabla \vec{\mathbf{c}}|_{\partial \Omega} = 0, \tag{6}$$

where $y$ is the state variable (*e.g.* concentration or current), $p$ is the pressure or potential field driving the flow, $\vec{\mathbf{c}}$ is the resulting velocity field, $\kappa$ is the permeability / permittivity, $s_q$ is the pressure/potential forcing term, $s_y$ is the rate of state variable injection into the system. The domain boundary is denoted by $\partial \Omega$ and the outward normal by $\vec{\mathbf{n}}$. The initial condition for tracer is zero over the entire domain.

In order to permit evaluation against ground truth, we used the permeability field in Fig. 1(a) based on a geologic model to study the flow of fluids through the earth subsurface under naturally and artificially induced pressure gradients. The data were generated by numerical simulation of eqn. (5) using a proprietary high-fidelity solver for $T = 12500\text{s}$ with spatially varying pressure loadings between $\pm 100$ units and with random temporal fluctuations (SNR of 20dB). Random amounts of tracer varying between 0 and 5 units were injected and concentration measured at 1s intervals at the 275 sites marked in the image. A video of the simulation is provided as supplemental to the manuscript, and the data and model are available on request . These concentration profiles at the 275 locations are used as the time-series data $\mathbf{y}$ input to the multi-scale graphical model of eqn. (1).

Estimation was done for $K = 20$, with multiple initializations and hyper-parameter selection as described above. The $K$-means step was initialized by distributing seed locations uniformly at random. The model orders $P$ and $Q$ were kept constant at 50 and 25 respectively. Labels and colors of the sites in Fig. 1(b) indicate the clusters identified by the $K$-means step for one initialization of the estimation procedure, while the estimated multi-scale graphical structure is shown in Figures 1(c)–(d). The global graphical structure (§Fig. 1(c)) correctly captures large-scale features in the ground truth. Furthermore, as seen in Fig. 1(d) the local graphical structure (given by the coefficients of B) are sparse and spatially compact. Importantly, the local graphs are spatially more contiguous than the initial $K$-means clusters and only approximately 40% of the labels are conserved between

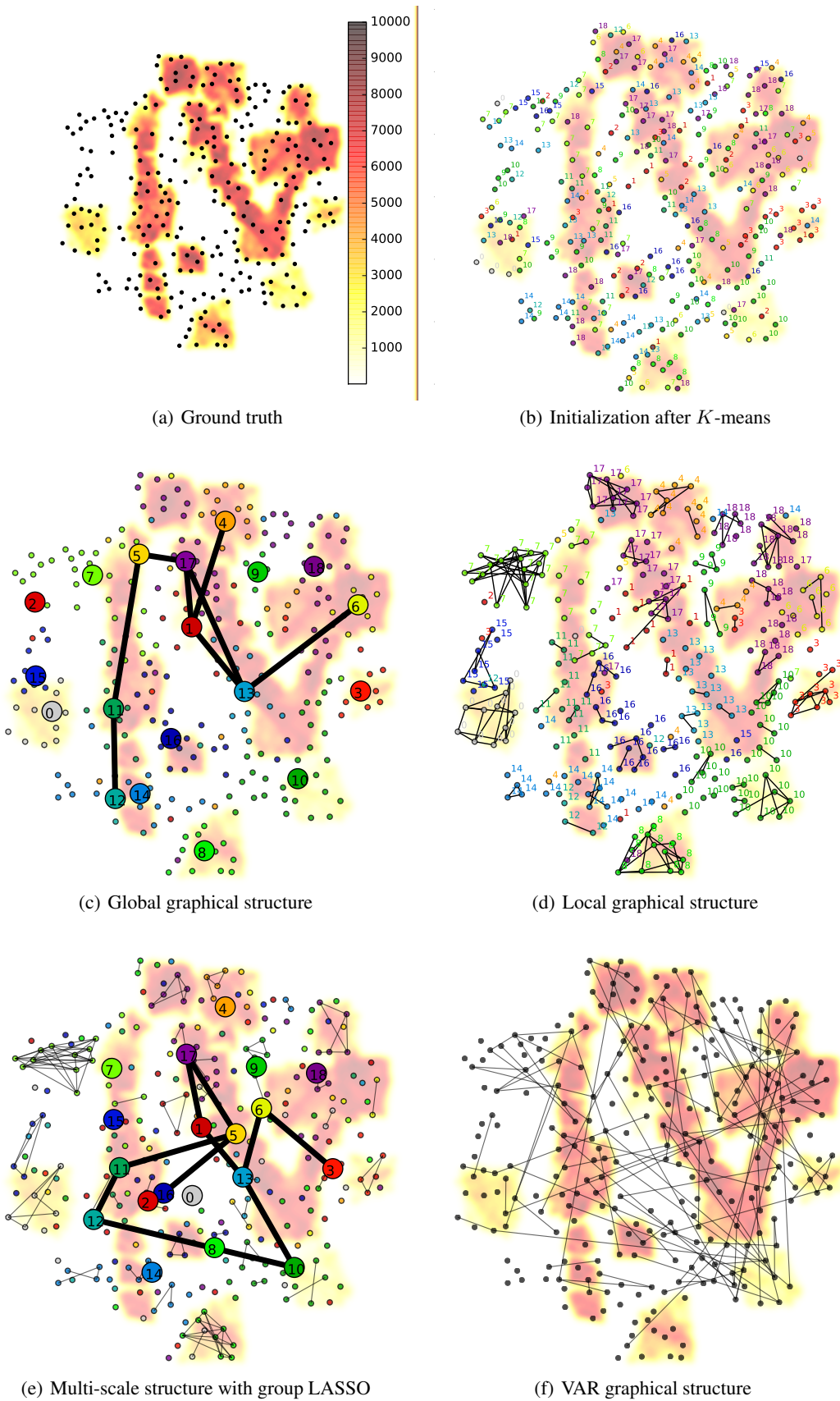

(a) Ground truth

(b) Initialization after $K$-means

(c) Global graphical structure

(d) Local graphical structure

(e) Multi-scale structure with group LASSO

(f) VAR graphical structure

Figure 1: **Fig.(a).** Ground truth permeability ($\kappa$) map overlaid with locations where the tracer is injected and measured. **Fig.(b).** Results of $K$−means initialization step. Colors and labels both indicate cluster assignments of the sites. **Fig.(c).** The global graphical structure for latent variable $\mathbf{x}$. The nodes are positioned at the centroids of the corresponding local graphs. **Fig.(d).** The local graphical structure. Again, colors and labels both indicate cluster (*i.e.* global component) assignments of the sites. **Fig.(e).** The multi-scale graphical structure obtained when the Gaussian function prior is replaced by group LASSO on A . **Fig.(f).** The graphical structure estimated using non-hierarchal VAR with group LASSO.

the $K$-means initialization and the final solution. Furthermore, as shown in Supplemental Appendix F, the estimated graphical structure is fairly robust to initialization, especially in recovering the global graph structure. For all initializations, estimation from a cold-start converged in 65–90 BCD iterations, while warm-starts converged in $< 5$ iterations.

Fig. 1(e) shows the results of estimating the multi-scale model when the penalty term of eqn. (3) for the C–D process prior is replaced by group LASSO. This result highlights the importance of the physically derived prior to reconstruct the graphical structure of the problem. Fig. 1(f) shows the graphical structure estimated using a non-hierarchal VAR model with group LASSO on the coefficients [11] and auto-regressive order $P = 10$. Firstly, this is a significantly larger model with $P \times N^2$ coefficients as compared $\mathcal{O}(P \times N) + \mathcal{O}(Q \times K^2)$ for the hierarchical model, and is therefore *much more* expensive to compute. Furthermore, the estimated graph is denser and harder to interpret in the terms of the underlying problem, with many long range edges intermixed with short range ones. In all cases, model hyper-

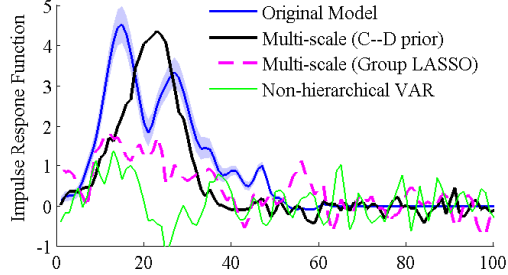

Figure 2: Response functions at node in cmpnt 17 to impulse in cmpnt 1 of Fig. 1(c). Plotted are the impulse responses for eqn. (5) along with 90% bands, the multi-scale model with C–D prior, the multi-scale model with group LASSO prior, and the non-hierarchical VAR model with group LASSO prior.

parameters were selected via 10-fold cross-validation described in Supplemental Appendix G. Interestingly, in terms of misfit (*i.e.* training ) error $\left( \sum_t \|y[t] - \hat{y}[t]\| \, / \sum_t \|y[t]\| \right)$ , the non-hierarchal VAR model performs best ($\approx \%12.1 \pm 4.4$ relative error) while group LASSO and C–D penalized hierarchal models perform equivalently ( $18.3 \pm 5.7\%$ and $17.6 \pm 6.2\%$) which can be attributed to the higher degrees of freedom available to non-hierarchical VAR. However, in terms of cross-validation (*i.e.* testing) error, the VAR model was the worst ( $94.5 \pm 8.9\%$) followed by group LASSO hierarchal model ($48.3 \pm 3.7\%$). The model with the C–D prior performed the best, with a relative-error of $31.6 \pm 4.5\%$.

To characterize the dynamics estimated by the various approaches, we compared the impulse response functions (IRF) of the graphical models with that of the ground truth model (§eqn. (5)). The IRF for a node $i$ is straightforward to generate for eqn. (5), while those for the graphical models are obtained by setting $\mathbf{v}_0[i] = 1$ and $\mathbf{v}_0[j] = 0$ for all $j \neq i$ and $\mathbf{v}_t = 0$ for $t > 0$ and then running their equations forward in time. The responses at a node in global component 17 of Fig. 1(c) to an impulse at a node in global component 1 is shown in Fig. 2. As the IRF for eqn. (5) depends on the driving pressure field which fluctuates over time, the mean IRF along with 90% bands are shown. It can be observed that the multi-scale model with the C–D prior is much better at replicating the dynamical properties of the original system as compared to the model with group LASSO, while a non-hierarchical VAR model with group LASSO fails to capture any relevant dynamics. The results of comparing IRFs for other pairs of sites were qualitatively similar and therefore omitted.

## 5   Conclusion

In this paper, we proposed a new approach that combines machine-learning / data-driven techniques with physically derived priors to reconstruct the connectivity / network structure of multi-scale spatio-temporal systems encountered in multiple fields such as exploration geophysics, atmospheric and ocean sciences . Simple yet computationally efficient algorithms for estimating the model were developed through a set of relaxations and regularization. The method was applied to the problem of learning the connectivity structure for a general class of problems involving flow through a permeable medium under pressure/potential fields and the advantages of this method over alternative approaches were demonstrated. Current directions of investigation includes incorporating different types of physics such as hyperbolic (*i.e.* wave) equations into the model. We are also investigating applications of this technique to learning structure in other domains such as brain networks, traffic networks, and biological and social networks.

## Footnotes

[1] http://clopinet.com/isabelle/Projects/NIPS2009+/

[2] The Péclet number $\mathrm{Pe} = Lc/\kappa$ is a dimensionless quantity which determines the ratio of advective to diffusive transfer, where $L$ is the characteristic length, $c$ is the advective velocity and $\kappa$ is the diffusivity of the system

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
