[Supplementary Material]

# Multi-scale Graphical Models for Spatio-Temporal Processes: Technical Supplement

**Firdaus Janoos**$^*$  **Huseyin Denli**  **Niranjan Subrahmanya**
ExxonMobil Corporate Strategic Research
Annandale, NJ 08801

## A  Proof for Theorem 1

*Proof.* The original model is given by:

$$\mathbf{x}[t] = \sum_{p=1}^{P} \mathrm{A}[p]\mathbf{x}[t-p] + \mathbf{u}[t], \tag{1}$$

$$\mathbf{y}[t] = \sum_{q=1}^{Q} \mathrm{B}[q]\mathbf{y}[t-q] + \mathrm{Z}\mathbf{x}[t] + \mathbf{v}[t]. \tag{2}$$

Define the short-hand notations $\mathbb{E}\left\{\cdot \mid t\right\} \triangleq \mathbb{E}\left\{\cdot \mid \mathbf{y}_p;\ s \le t\right\}$ and $\hat{\mathrm{B}}_q \triangleq (\mathrm{Z}\mathrm{Z}^\top) \odot \mathrm{B}_p$, captures the constraint that site $i \leftrightarrow$ site $j$ if they belong to the same global component $k$. Also to reduce clutter, we use subscripts for time indexing. Suitably adjusting the time-indices, the model of eqn. (1) can be re-written as:

$$\mathbf{x}_{t+1} = \sum_{p=0}^{P-1} \mathrm{A}_p\mathbf{x}_{t-p} + \mathbf{u}_t \tag{3}$$

$$\mathbf{y}_t = \sum_{q=1}^{Q} \hat{\mathrm{B}}_q\mathbf{y}_{t-q} + \mathrm{Z}\mathbf{x}_t + \mathbf{v}_t, \tag{4}$$

It is assumed that the model is stable, that is all the roots of $\det\left|\mathrm{I} - \sum_p \mathrm{A}_p z^{-p}\right|$ (*i.e.* $z$–transform of A) lie within the unit circle $|z| < 1$ system. In order to prove Theorem 1 we first introduce the following proposition:

**Proposition 1.** *Assuming that the system is stable, , then eqn.* (3) *conditioned on data is equivalent to:*

$$\mathbf{x}_{t+1 \mid t} = \sum_{p=0}^{P-1} \mathrm{A}_p\mathbf{x}_{t-p \mid t-p-1} + \sum_{p=0}^{P-1} \mathrm{G}_p\epsilon_{t-p} \tag{5}$$

$$\mathbf{y}_t = \sum_{q=1}^{Q} \hat{\mathrm{B}}_q\mathbf{y}_{t-q} + \mathrm{Z}\mathbf{x}_{t \mid t-1} + \epsilon_t, \tag{6}$$

*where* $\mathbf{x}_{t \mid t-1} = \mathbb{E}\left\{\mathbf{x}_t \mid t-1\right\}$, $\mathbf{y}_{t \mid t-1} = \mathbb{E}\left\{\mathbf{y}_t \mid t-1\right\}$, $\epsilon_t = y_t - \mathbf{y}_{t \mid t-1}$, *and* $\mathrm{G}$ *are* $K \times N$ *matrices defined as follows:*

$$\mathrm{G}_p = \sum_{s=p}^{P-1} \mathrm{A}_s \mathbb{C}ov\left\{\mathbf{x}_{t-s}\epsilon_t^\top\right\}\Sigma_\epsilon^{-1} \qquad for\ p = 0 \dots P-1$$

$^*$Corresponding Author. firdaus@ieee.org

Proof for this proposition is in Appendix A.1. Defining $\hat{\mathbf{X}}, \mathbf{Y}$ and $\Upsilon$ as the $z$–transform of $\hat{\mathbf{x}}, \mathbf{y}$ and $\epsilon$ respectively, we get the $z$–transform of eqn. (5):

$$\left(\mathrm{I} - \sum_{q=0}^{Q-1} \hat{\mathrm{B}}_q z^{-p}\right) \mathbf{Y} = \left(\mathrm{I} + \mathrm{Z}\left(\mathrm{I} - \sum_{p=0}^{P-1} \mathrm{A}_p z^{-p}\right)^{-1}\left(\sum_{p=0}^{P-1} \mathrm{G}_p \mathbf{z}^{-p}\right)\right) \Upsilon$$

As, Z is a $N \times K$ matrix of rank $K$ the left pseudo-inverse of Z is well defined and :

$$\left(\mathrm{I} - \sum_{p=0}^{P-1} \mathrm{A}_p z^{-p}\right)\mathrm{Z}^+\left(\mathrm{I} - \sum_{q=0}^{Q-1}\hat{\mathrm{B}}_q z^{-q}\right)\mathbf{Y} = \left(\sum_{p=0}^{P-1}\mathrm{G}_p \mathbf{z}^{-p}\right)\Upsilon + \left(\mathrm{I} - \sum_{p=0}^{P-1}\mathrm{A}_p z^{-p}\right)\mathrm{Z}^+\Upsilon$$

therefore

$$Z^* - \sum_{p=0}^{P-1}\mathrm{A}_p\mathrm{Z}^+ z^{-p} - \sum_{q=0}^{Q-1}\mathrm{Z}^+\hat{\mathrm{B}}_q z^{-q} + \sum_{p=0}^{P-1}\sum_{q=0}^{Q-1}\mathrm{A}_p\mathrm{Z}^+\hat{\mathrm{B}}_q z^{-(p+q)}$$

$$= \left(\sum_{p=0}^{P-1}\mathrm{G}_p\mathbf{z}^{-p}\right)\Upsilon + \left(\mathrm{I} - \sum_{p=0}^{P-1}\mathrm{A}_p z^{-p}\right)\mathrm{Z}^+\Upsilon$$

Assuming that the system is minimal[3] that is there is strictly no smaller equivalent model, this represents an $N \times N$ rank-$K$ system of auto-regressive order $P + Q$ and moving-average order $P$. $\square$

## A.1 Proof for Proposition 1

*Proof.* Firstly, because the assumption on the roots of $\det\left|\mathrm{I} - \sum_p \mathrm{A}_p z^p\right|$, the system is wide sense stationary (WSS) [1]. Now, by the projection property of linear systems the residual $\epsilon_t \perp \mathbf{y}_p$; $s < t$ and therefore:

$$\mathbb{E}\left\{\cdot \mid t\right\} = \mathbb{E}\left\{\cdot \mid t-1\right\} + \mathbb{E}\left\{\cdot \mid \epsilon_t\right\}$$

Therefore, the conditional estimate of $\mathbf{x}_t$ :

$$\mathbf{x}_{t+1 \mid t} = \sum_{p=0}^{P-1}\mathbb{E}\left\{\mathrm{A}_p\mathbf{x}_{t-p} + \mathbf{u}_t \mid t\right\}$$

$$= \sum_{p=0}^{P-1}\mathrm{A}_p\mathbf{x}_{t-p \mid t},$$

using the property that $\mathbf{u}_t$ independent on $\mathbf{y}_0\ldots\mathbf{y}_t$. Defining $\mathrm{H}_p = \mathbb{C}\mathrm{ov}\left\{\mathbf{x}_{t-p}\epsilon_t^\top\right\}\Sigma_\epsilon^{-1}$ and because of the WSS condition, it does not depend on $t$. Therefore,

$$\mathbf{x}_{t+1 \mid t} = \sum_{p=0}^{P-1}\mathrm{A}_p\left(\mathbf{x}_{t-p \mid t-1} + \mathbb{E}\left\{\mathbf{x}_{t-p} \mid \epsilon_t\right\}\right)$$

$$= \mathrm{A}_0\mathbf{x}_{t \mid t-1} + \sum_{p=1}^{P-1}\mathrm{A}_p\mathbf{x}_{t-p \mid t-1} + \sum_{p=0}^{P-1}\mathrm{A}_p\mathrm{H}_p\epsilon_t$$

$$= \mathrm{A}_0\mathbf{x}_{t \mid t-1} + \sum_{p=1}^{P}\mathrm{A}_p\left(\mathbf{x}_{t-p \mid t-2} + \mathbb{E}\left\{\mathbf{x}_{t-p} \mid \epsilon_{t-1}\right\}\right) + \sum_{p=0}^{P-1}\mathrm{A}_p\mathrm{H}_p\epsilon_t$$

$$= \mathrm{A}_0\mathbf{x}_{t \mid t-1} + \mathrm{A}_1\mathbf{x}_{t-1 \mid t-2} + \sum_{p=2}^{P}\mathrm{A}_p\mathbf{x}_{t-p \mid t-2} + \sum_{p=1}^{P}\mathrm{A}_p\mathrm{H}_{p-1}\epsilon_{t-1} + \sum_{p=0}^{P-1}\mathrm{A}_p\mathrm{H}_p\epsilon_t$$

Continuing this expansion, and setting $G_p = \sum_{s=p}^{P-1} A_s H_{s-p}$ for $p = 0 \ldots P - 1$, we get:

$$
\begin{aligned}
\mathbf{x}_{t+1 \mid t} &= \sum_{p=0}^{P-1} A_p \mathbf{x}_{t-p \mid t-p-1} + \sum_{p=0}^{P-1} \left[ \sum_{s=p}^{P-1} A_s H_{s-p} \right] \epsilon_{t-p} \\
&= \sum_{p=0}^{P-1} A_p \mathbf{x}_{t-p \mid t-p-1} + \sum_{p=0}^{P-1} G_p \epsilon_{t-p}
\end{aligned}
\tag{7}
$$

Moreover, since:

$$
\mathbf{y}_{t \mid t-1} = \mathbb{E} \left\{ \sum_{q=1}^{Q} \hat{B}_q \mathbf{y}_{t-q} + Z \mathbf{x}_t + \mathbf{v}_t \mid t-1 \right\} = \sum_{q=1}^{Q} \hat{B}_q \mathbf{y}_{t-q} + Z \mathbf{x}_{t \mid t-1}
$$

we get the following innovations model:

$$
\mathbf{y}_t = \sum_{q=1}^{Q} \hat{B}_q \mathbf{y}_{t-q} + Z \mathbf{x}_{t \mid t-1} + \epsilon_t.
\tag{8}
$$

Defining $\eta_t = \mathbf{x}_t - \mathbf{x}_{t \mid t-1}$, we see that:

$$
\epsilon_t = y_t - \mathbf{y}_{t \mid t-1} = Z \mathbf{x}_t + \mathbf{v}_t - Z \mathbf{x}_{t \mid t-1} = Z \eta_t + \mathbf{v}_t
\tag{9}
$$

while:

$$
\begin{aligned}
\eta_{t+1} &= \sum_{p=0}^{P-1} A_p \left( \mathbf{x}_{t-p} - \mathbf{x}_{t-p \mid t-1} \right) + \mathbf{u}_t - \sum_{p=0}^{P-1} G_p \epsilon_{t-p} \\
&= \sum_{p=0}^{P-1} A_p \eta_{t-p} + \mathbf{u}_t - \sum_{p=0}^{P-1} G_p \left( Z \eta_{t-p} + \mathbf{v}_{t-p} \right) \\
&= \sum_{p=0}^{P-1} \left( A_p - G_p Z \right) \eta_{t-p} + \mathbf{u}_t - \sum_{p=0}^{P-1} G_p \mathbf{v}_{t-p}
\end{aligned}
$$

$\square$

# B  Proof for Theorem 2

*Proof.* In order to prove the theorem, we introduce the following proposition:

**Proposition 2.** *For a 1–d C–D system with infinite boundary conditions and constant Péclet number, for an impulse at the origin $x = 0$, the response in the near-field (i.e. at a point close to the origin) can be approximated by $G(t) \approx \delta(t)$, the Dirac delta function, while in far–field (i.e. at a point far from the origin) and be approximated by a Gaussian function: $G(t) \approx \exp\{-0.5(t - \mu^2 \sigma^{-2})\} / \sqrt{2\pi\sigma^2}$ up to a multiplicative factor of order $\exp\{-\mathcal{O}(t^3)\}$, where $\mu$ is equal to the distance and $\sigma^2$ is proportional to the product of the distance and the Péclet number.*

The proof to Proposition 2 is given Appendix B.1. Therefore, for the system of Theorem 2, the response at location $\mathbf{x}_i$ to an impulse at $\mathbf{x}_j = 0$ can be approximated by a Gaussian function: $G(t)_{i,j} = \mathcal{N}\left(\mu_{i,j}, \sigma_{i,j}^2\right) = \exp\{-0.5(t - \mu_{i,j})^2 \sigma_{i,j}^{-2}\} / \sqrt{2\pi\sigma_{i,j}^2}$, where $\mu_{i,j}$ is equal to the distance between $\mathbf{x}_i$ and $\mathbf{x}_j$ and $\sigma_{i,j}$ is proportional to the production of the distance and the Péclet number. Moreover, the response at $\mathbf{x}_i$ to an impulse at $\mathbf{x}_i$ can be approximated by a Dirac delta. Therefore, in Fourier domain the *transfer function* of the $2 \times 2$ system consisting of only nodes $\mathbf{x}_i$ and $\mathbf{x}_j$:

$$
\hat{\Psi}(\omega) = \begin{bmatrix} 1 & \widehat{G}_{i,j}(\omega) \\ \widehat{G}_{j,i}(\omega) & 1 \end{bmatrix}, \qquad \text{where} \qquad \Psi(t) = \begin{bmatrix} \delta(t) & G_{i,j}(t) \\ G_{j,i}(t) & \delta(t) \end{bmatrix}
\tag{10}
$$

is the impulse response matrix of the $2 \times 2$ system consisting of only nodes $\mathbf{x}_i$ and $\mathbf{x}_j$, and $\hat{\Psi}(\omega)$ is Fourier transform (FT) of $\Psi[t]$. Also, $\widehat{G}_{i,j}(\omega) \approx \pi^{-\frac{1}{2}} \exp\{-i\omega\mu_{i,j}\} \exp\{-\sigma_{i,j}^2\omega^2\} \otimes \mathcal{F}(\omega)$ is the FT of $G[t]_{i,j}$ while the FT of $\delta(t)$ is 1. The FT of approximation to $\widehat{G}_{i,j}(\omega)$ is obtained by convolution of the FT of the multiplicative error term $\exp\left\{-\mathcal{O}(t^3)\right\}$ with the FT of $\exp\left\{-0.5(t - \mu^2\sigma^{-2})\right\}/\sqrt{2\pi\sigma^2}$.

However for stable VAR system, the transfer function between any pair of variables conditioned on the rest is[1]:

$$\hat{\Psi}(\omega) = \begin{bmatrix} \hat{A}_{i,i}(\omega) & \hat{A}_{i,j}(\omega) \\ \hat{A}_{j,i}(\omega) & \hat{A}_{j,j}(\omega) \end{bmatrix}^{-1}, \tag{11}$$

where $\hat{A}_{i,j}(\omega)$ is the FT of $A_{i,j}[t]$. Inverting the matrix in eqn. (10) and equating with terms of eqn. (11) gives

$$\hat{A}_{i,j}(\omega) = -\frac{\widehat{G}_{i,j}(\omega)}{1 - \widehat{G}(\omega)_{i,j}^* \widehat{G}(\omega)_{i,j}}$$

Taking logs of the absolute value (squared) yields

$$\log(\hat{A}_{i,j}(\omega)^* \hat{A}_{i,j}(\omega)) = \log\left(\widehat{G}_{i,j}(\omega)^* \widehat{G}_{i,j}(\omega)\right) - 2\log\left(1 - \widehat{G}_{i,j}(\omega)^* \widehat{G}_{i,j}(\omega)\right).$$

However, since $|\widehat{G}| \ll 1$ the square magnitude $\widehat{G}_{i,j}(\omega)^* \widehat{G}_{i,j}(\omega) \ll 1$ which implies that

$$\log\left(1 - \widehat{G}_{i,j}(\omega)^* \widehat{G}_{i,j}(\omega)\right) \approx 0 + \mathcal{O}\left(\widehat{G}_{i,j}(\omega)^* \widehat{G}_{i,j}(\omega)\right)$$

Substituting gives

$$\log(\hat{A}_{i,j}(\omega)^* \hat{A}_{i,j}(\omega)) \approx \log\left(\widehat{G}_{i,j}(\omega)^* \widehat{G}_{i,j}(\omega)\right) + \mathcal{O}\left(\widehat{G}_{i,j}(\omega)^* \widehat{G}_{i,j}(\omega)\right)$$

which implies that the FT of $|\hat{A}_{i,j}(\omega)| \approx |\widehat{G}_{i,j}(\omega)|$ and therefore $A[t]_{i,j}$ can be approximated by a Gaussian function $\mathcal{N}(\mu_{i,j}, \sigma_{i,j}^2)$. Moreover the approximation has a multiplicative error of order $\exp\left\{-\mathcal{O}(t^3)\right\}$.

□

## B.1 Proof for Proposition 2

Consider the dimensionless constant-coefficient convection-diffusion equation in 1–d:

$$\frac{\partial f}{\partial t} + \frac{\partial f}{\partial x} = \gamma\frac{\partial^2 f}{\partial t^2},$$

where $f$ is the process, $x$ is the spatial coordinate and $\gamma = 1/\text{Pe}$ is the inverse of the Péclet number of the system. Under infinite boundary conditions, the Green's function (*i.e.* impulse response) has the form

$$g(x,t) = \frac{1}{\sqrt{4\pi\gamma t}} \exp\left\{-\frac{1}{2}\frac{(x-t)^2}{2\gamma t}\right\}. \tag{12}$$

In order to derive an approximation, assume $\gamma = 1$ without loss of generality.

$$g(x,t) = \frac{1}{\sqrt{2\pi}} \exp\left\{-\frac{1}{2}\frac{(x-t)^2}{2t} - \frac{1}{2}\log 2t\right\} \tag{13}$$

Writing $t = x + \tau$, a Taylor series expansion (TSE) of the term in the exponent gives:

$$\exp\left\{-\frac{1}{2}\frac{\tau^2}{2(x+\tau)} - \frac{1}{2}\log 2(x+\tau)\right\}$$

$$= \exp\left\{-\frac{1}{2}\frac{\tau^2}{2}\left((2x)^{-1} - (2x)^{-2}\tau + (2x)^{-3}\tau^2 - (2x)^{-4}\tau^3 + \ldots\right)\right.$$

$$\left. - \frac{1}{2}\left(\log 2x + (2x)^{-1}\tau - \frac{(2x)^{-2}}{2}\tau^2 + \frac{(2x)^{-3}}{3}\tau^3 + \ldots\right)\right\}$$

$$= \exp\left\{-\frac{1}{2}\frac{\tau^2}{2x} - \frac{1}{2}\log 2x - \frac{1}{2}\left(\tau^3\left((2x)^{-1} - (2x)^{-2}\right) - \tau^4\left(\frac{(2x)^{-2}}{2} - (2x)^{-3}\right) + \ldots\right)\right\}$$

$$\approx \frac{1}{\sqrt{x}}\exp\left\{-\frac{1}{2}\frac{\tau^2}{2x}\right\}\exp\left\{-\frac{1}{2}\mathcal{O}(\tau^3)\right\}.$$

Therefore, for the far-field (*i.e.* $x \gg 0$) the approximation holds because:

$$\frac{\exp\left\{-\frac{1}{2}\frac{\tau^2}{2x}\right\}}{\exp\left\{-\frac{1}{2}\mathcal{O}(\tau^3)\right\}} \to 1 \qquad \text{as} \qquad \tau \to 0$$

$$\text{and} \qquad \exp\left\{-\frac{1}{2}\tau^3\right\} - \exp\left\{-\frac{1}{2}\frac{\tau^2}{2x}\right\} \to 0 \qquad \text{as} \qquad \tau \to \infty,$$

And for the near-field *i.e.* as $x \to 0$ the response function $g(x,t) \to \delta(t)$.

## C  Proof for Theorem 3

*Proof.* We start by introducing the following notation - let $\mathbf{y}_i \in \mathbf{x}_k$ imply that $Z_{i,k} = 1$. For two vertices $\mathbf{y}_i$ and $\mathbf{y}_j$, let $\mathbf{y}_i \sim \mathbf{y}_j$ indicate that there is at least one shared latent component $\mathbf{y}_i \in \mathbf{x}_k$ and $\mathbf{y}_j \in \mathbf{x}_k$, *i.e.* $Z_{i,k}Z_{j,k} = 1$. Also, let $\hat{B} = (ZZ^\top) \odot B$, where $\odot$ is the Hadamard product. Therefore, $\hat{B}_{i,j} = 0 \Rightarrow \exists k$ s.t. $\mathbf{y}_i \sim \mathbf{y}_j$. Without loss of generality, we will prove this assertion for the case below, that is:

**Proposition 3.** *For a given* Z, *the least-squares (LS) local optimum* $A^*$ *and* $\mathbf{x}^*$ *to*

$$\mathbf{x}[t] = \sum_{p=1}^{P} A[p]\mathbf{x}[t-p] + \mathbf{u}[t] \tag{14}$$

$$\mathbf{y}[t] = \hat{B}\mathbf{y}[t-1] + Z\mathbf{x}[t] + \mathbf{v}[t], \tag{15}$$

*is also a local optimum for*

$$\mathbf{x}[t] = \sum_{p=1}^{P} A[p]\mathbf{x}[t-p] + \mathbf{u}[t] \tag{16}$$

$$\mathbf{y}[t] = C\mathbf{y}[t-1] + Z\mathbf{x}[t] + \mathbf{v}[t], \tag{17}$$

*for some diagonal matrix* C.

The more general case of Theorem 3 is a straightforward extension of this proof.

First, we observe that as eqn. (14) is decoupled from B, the local minimum $A^*[p]$ conditioned on $\mathbf{x}^*$ is de-coupled from B. Therefore, we only need to show that if

$$\mathbf{x}^+ = \underset{\mathbf{x}}{\operatorname{argmin}} f(\mathbf{x})$$

where:

$$f(\mathbf{x}) = \sum_t \left[\mathbf{y}[t] - \hat{\mathbf{B}}\mathbf{y}[t-1] - \mathbf{Z}\mathbf{x}[t]\right]^\top [\mathbf{y}[t] - \mathbf{B}\mathbf{y}[t-1] - \mathbf{Z}\mathbf{x}[t]]$$

then $\mathbf{x}^+$ also a LS solution to eqn. (17).

Now,

$$\frac{1}{2}\nabla_{\mathbf{x}[t]}f(\mathbf{x}) = \left[\mathbf{y}[t] - \mathbf{B}\mathbf{y}[t-1] - \mathbf{Z}\mathbf{x}[t]\right]^\top \mathbf{Z} = \mathbf{y}[t]^\top \mathbf{Z} - \mathbf{y}[t-1]^\top \hat{\mathbf{B}}^\top \mathbf{Z} - \mathbf{x}[t]^\top \mathbf{Z}^\top \mathbf{Z}$$

Defining $\eta = \hat{\mathbf{B}}\mathbf{y}[t-1]$, we get $\eta_i = \left[\hat{\mathbf{B}}\mathbf{y}[t-1]\right]_i = \sum_j \mathbf{y}_j[t-1]\hat{\mathbf{B}}_{i,j}$, that is $\eta_i = \sum_j \mathbf{B}_{i,j}\mathbf{y}_j[t-1]$ for all $\mathbf{y}_j$ such that $\mathbf{y}_i \sim \mathbf{y}_j$.

Moreover, $\left[\eta^\top \mathbf{Z}\right]_k = \sum_i \eta_i \mathbf{Z}_{i,k}$ that is $\left[\eta^\top \mathbf{Z}\right]_k = \sum_i \eta_i$ for all $\mathbf{y}_i \in \mathbf{x}_k$. Therefore, chaining these two we get $\left[\mathbf{Z}^\top \eta\right]_k = \sum_i \sum_j \mathbf{B}_{i,j}\mathbf{y}_j[t-1]$ for all $i$ s.t. $\mathbf{y}_i \in \mathbf{x}_k$ and $j$ s.t. $\mathbf{y}_i \sim \mathbf{y}_j$, which is the same as $\left[\mathbf{Z}^\top \eta\right]_k = \sum_{\mathbf{y}_i \in \mathbf{x}_k} \sum_{\mathbf{y}_j \in \mathbf{x}_k} \mathbf{B}_{i,j}\mathbf{y}_j[t-1]$.

Therefore

$$\frac{1}{2}\left[\nabla_{\mathbf{x}[t]}f(\mathbf{x})\right]_k = \left[y[t]^\top \mathbf{Z}\right] - \left[\mathbf{Z}^\top \eta\right]_k = \sum_{\mathbf{y}_i \in \mathbf{x}_k} \sum_{\mathbf{y}_j \in \mathbf{x}_k} \mathbf{B}_{i,j}\mathbf{y}_j[t-1] - \left[\mathbf{x}[t]^\top \mathbf{Z}^\top \mathbf{Z}\right]_k \qquad (18)$$

This gradient is equivalent to $\frac{1}{2}\nabla_{\mathbf{x}[t]}g(\mathbf{x})$ where $g(\mathbf{x})$ is the LS objective of eqn. (17).

$$\frac{1}{2}\nabla_{\mathbf{x}[t]}g(\mathbf{x}) = \mathbf{y}[t]^\top \mathbf{Z} - \mathbf{y}[t-1]^\top \mathbf{C}\mathbf{Z} - \mathbf{x}[t]^\top \mathbf{Z}^\top \mathbf{Z}$$

where $\left[\mathbf{y}[t-1]^\top \mathbf{C}\mathbf{Z}\right]_k = \sum_{j \in \mathbf{x}_k} \mathbf{C}_j\mathbf{y}_j[t-1]$, for $\mathbf{C}_j = \sum_{\mathbf{y}_i \in \mathbf{x}_k} \mathbf{B}_{i,j}$.

Note that the Hessian of the objective function is independent of B and C. Therefore, the two quadratic problems differ only in a constant independent of $\mathbf{x}$ and therefore have the same optimal solutions.

$\square$

# D  Optimization with respect to x

The state-space model

$$\mathbf{x}[t] = \sum_{p=1}^{P} \mathbf{A}[p]\mathbf{x}[t-p] + \mathbf{u}[t] \qquad \text{and} \qquad \mathbf{y}[t] = \sum_{q=1}^{Q} \mathbf{C}[p]\mathbf{y}[t-q] + \mathbf{Z}\mathbf{x}[t] + \mathbf{v}[t], \qquad (19)$$

can be re-written as the followed augmented state space form

$$\zeta[t+1] = \mathbf{A}_{\text{aug}}\zeta'[t] + \nu[t] \qquad \text{and} \qquad \vartheta[t] = \mathbf{Z}_{\text{aug}}\zeta'[t] + \mathbf{v}[t],$$

where $\zeta[t] = (\mathbf{x}[t] \ldots \mathbf{x}[t-P])^\top$ is the augmented state, $\vartheta[t] = \left(\mathbf{y}[t] - \sum_{q=1}^{Q} \mathbf{C}^{(n)}[q]\mathbf{y}[t-q]\right)^\top$ is the augmented observation, $\nu[t] = (\mathbf{u}[t], 0 \ldots 0)^\top$ is the augmented state innovations the matrices $\mathbf{A}_{\text{aug}}$ is a $PK \times PK$ matrix with $\mathbf{A}^{(n)}[1] \ldots \mathbf{A}^{(n)}[P]$ on the first rowand $\mathbf{Z}_{\text{aug}}$ is a $N \times KP$ matrix constructed from $\mathbf{A}^{(n)}$ and $\mathbf{Z}^{(n)}$. Namely:

$$\mathbf{A}_{\text{aug}} = \begin{bmatrix} \mathbf{A}[1] & \mathbf{A}[2] & \ldots & \mathbf{A}[P] \\ \mathbf{I} & 0 & \ldots & 0 \\ 0 & \mathbf{I} & \ldots & 0 \\ \vdots & \vdots & \ddots & \vdots \end{bmatrix} \qquad \mathbf{Z}_{\text{aug}} = [\mathbf{Z} \quad 0 \quad \ldots \quad 0]$$

Rewriting the summation of time in vector-matrix format for notational clarity, the optimization problem is then:

$$\zeta^{(n+1)} = \underset{\zeta}{\operatorname{argmin}} \left\| \vartheta - \Delta_{\text{aug}} \zeta \right\|^2 + \lambda_0 \left\| \Lambda_{\text{aug}} \zeta \right\|^2$$

where $\Delta$ and $\Lambda$ are the $T \times T$ block-diagonal matrices:

$$\Delta = \begin{bmatrix} Z_{\text{aug}} & 0 & \dots & 0 \\ 0 & Z_{\text{aug}} & \dots & 0 \\ \vdots & \vdots & \ddots & \vdots \\ 0 & 0 & \dots & Z_{\text{aug}} \end{bmatrix} \qquad \Lambda_{\text{aug}} = \begin{bmatrix} -A_{\text{aug}} & I & 0 & \dots & 0 \\ 0 & -A_{\text{aug}} & I & \dots & 0 \\ \vdots & \vdots & \vdots & \ddots & \vdots \\ 0 & 0 & 0 & \dots & -A_{\text{aug}} \end{bmatrix}$$

This is a $T \times T$ tri-diagonal Toeplitz system with blocks of size $KP$ and which can be solved using specialized solvers that have running time of $\mathcal{O}(T \times KP^3)$ [2]. In case the system is ill-conditioned or rank-deficient, a small perturbation may be added along the diagonal of the matrix. However in practice, we have observed the system to be well conditioned, especially when close to a solution point.

## E    Proximal operators

### E.1    Operator for $Z$

Defining $Z_{i,\cdot} = \{Z_{i,1} \dots Z_{i,K}\}$, the proximal operator $\mathbf{prox}_h(Z^{(n)})$ with respect to $Z$, and writing the unit-ball constraint $\|Z_{i,\cdot}\|_2 \leq 1$ as an indicator function $\mathbb{I}_1 \left( \|Z_{i,\cdot}\|_2 \right)$, we get:

$$\mathbf{prox}_g(\hat{Z}) = \underset{Z}{\operatorname{argmin}} \lambda_3 \|Z\|_1 + \sum_{i=1}^{N} \mathbb{I}_1 \left( \|Z_{i,\cdot}\|_2^2 \right) + \frac{1}{2} \left\| Z - \hat{Z} \right\|^2$$

$$= \lambda_3 \sum_{i=1}^{N} \|Z_{i,\cdot}\|_1 + \sum_{i=1}^{N} \mathbb{I}_1 \left( \|Z_{i,\cdot}\|_2^2 \right) + \frac{1}{2} \sum_{i=1}^{N} \left\| Z_{i,\cdot} - \hat{Z})_{i,\cdot} \right\|^2$$

As the problem is decomposable into a sum of problems over $Z[1, \cdot] \dots Z[N, \cdot]$, consider the individual problem of the form:

$$\underset{\mathbf{z}}{\operatorname{argmin}} \lambda_3 \|\mathbf{z}\|_1 + \frac{1}{2} \|\mathbf{z} - \hat{\mathbf{z}})\|^2$$
$$\text{subject to} \quad \|\mathbf{z}\|_2 \leq 1$$

where $\mathbf{z} = \{\mathbf{z}_1 \dots \mathbf{z}_K\}$ represents any $Z_{i,\cdot}$.

Introducing Lagrange multipliers $\mu \in \mathbb{R}^+$ and $\eta \in \mathbb{R}^{+K}$, the dual is:

$$\mathcal{L}(\mathbf{z}, \mu, \eta) = \lambda_3 \|\mathbf{z}\|_1 + \frac{1}{2} \|\mathbf{z} - \hat{\mathbf{z}})\|^2 + \frac{\mu}{2} \left( \|\mathbf{z}\|_2^2 - 1 \right)$$

Dual feasibility implies:

$$0 = \partial \mathcal{L}(\mathbf{z}, \mu, \eta) \mathbf{z}_k = \lambda_3 \partial_k \|\mathbf{z}\|_1 + \mathbf{z}_k - \hat{\mathbf{z}}_k + \mu \mathbf{z}_k$$

therefore in vector format

$$\mathbf{z}(1 + \mu) = \hat{\mathbf{z}} - \lambda_3 \partial \|\mathbf{z}\|_1$$

where the sub-gradient

$$\partial_k \|\mathbf{z}\|_1 = \begin{cases} 1 & \text{if} \quad {}_k > 0 \\ -1 & \text{if} \quad \mathbf{z}_k < 0 \\ (-1, +1) & \text{if} \quad \mathbf{z}_k = 0 \end{cases}$$

Now if there is at-least one $k$ such that $|\hat{\mathbf{z}}_k| > \lambda$, then the following argument applies;

If $\mathbf{z}_k^* > 0$ then $\partial \|\mathbf{z}\|_1 = 1$. Therefore, $(1 + \mu)\mathbf{z}_k^* = (\mathbf{z}' - \lambda) > 0$. This implies that if $\mathbf{z}' > \lambda$ then $(1 + \mu)\mathbf{z}_k^* = (\hat{\mathbf{z}} - \lambda)$. Similarly if $\hat{\mathbf{z}} > -\lambda$ then $(1 + \mu)\mathbf{z}_k^* = (\hat{\mathbf{z}} + \lambda)$, and if $-\lambda \leq \hat{\mathbf{z}} \leq \lambda$ then $\mathbf{z}_k^* = 0$.

Defining $T_\lambda$ as the element-wise shrinkage operator, we get $(1 + \mu)\mathbf{z}^* = T_\lambda(\mathbf{z}')$. Moreover, setting $\mu = \min\left(\|T_\lambda(\mathbf{z}'\|_2 - 1, 0\right)$ satisfies the complementary slackness condition:

$$\mu^* > 0 \qquad \text{if} \qquad \|\mathbf{z}\|_2^2 = 1$$
$$\mu^* = 0 \qquad \text{if} \qquad \|\mathbf{z}\|_2^2 < 1$$

However if $|\hat{\mathbf{z}}_k| \leq \lambda$, then $\mathbf{z}^* = 0$ is the only feasible solution.

Therefore, the final solution is:

$$\mathbf{z}^* = \max\left(1, \frac{1}{\|T_\lambda(\hat{\mathbf{z}})\|_2}\right) T_\lambda(\hat{\mathbf{z}})$$

### E.2 Operator for A

The proximal operator for A is

$$\mathbf{prox}_g(\hat{A}) = \underset{A}{\arg\min} \sum_{i,j=1}^{K} \left[\lambda_1 \|D(\gamma_{i,j})A_{i,j}\|_2 + \frac{1}{2}\left\|A_{i,j} - \widehat{A_{i,j}}\right\|^2\right],$$

subject to $0 \leq \sum_p A_{i,j}[p] \leq 1$ for all $i, j = 1 \ldots K$, where $D(\gamma_{i,j})$ represents the operator $\widehat{\partial}_t + \gamma_{i,j}(p - \mu_{i,j})$ and is a $P \times P$ full-rank matrix. Since this can be split across all $A_{i,j}$, consider one problem of the form:

$$\mathbf{prox}_g(\hat{\mathbf{a}}) = \underset{\mathbf{a}}{\arg\min} \, \lambda_1 \|D\mathbf{a}\|_2 + \frac{1}{2}\|\mathbf{a} - \hat{\mathbf{a}}\|^2, \tag{20}$$

subject to $0 \leq \mathbb{1}^\top \mathbf{a} \leq 1$ where $\mathbf{a}$ represents any $A_{i,j}$, and $D$ represents the corresponding $D(\gamma_{i,j})$.

The Lagrangian of the problem is:

$$\mathcal{L}_{\mathbf{prox}_g}(\mathbf{a}, \eta_1, \eta_2) = \lambda_1 \|D\mathbf{a}\|_2 + \frac{1}{2}\|\mathbf{a} - \hat{\mathbf{a}}\|^2 - \eta_1 \mathbb{1}^\top \mathbf{a} + \eta_2(\mathbb{1}^\top \mathbf{a} - 1) \tag{21}$$

where $\eta_1 > 0$ and $\eta_2 > 0$ are Lagrange multipliers for $-\mathbb{1}^\top \mathbf{a} \leq 0$ and $\mathbb{1}^\top \mathbf{a} \leq 1$

As $D$ is full rank, define $\mathbf{b} = D\mathbf{a}$ and $\hat{\mathbf{b}} = D^{-1}\hat{\mathbf{a}}$. Therefore, eqn. (22) can be rewritten as :

$$\mathbf{prox}_g(\hat{\mathbf{a}}) = D\left[\underset{\mathbf{b}}{\arg\min} \, \lambda_1 \|\mathbf{b}\|_2 + \frac{1}{2}\left\|(\mathbf{b} - \hat{\mathbf{b}})\right\|^2\right],$$

subject to $0 \leq \mathbb{1}^\top D^{-1}\mathbf{b} \leq 1$.

The dual feasible solution $\mathbf{b}^*(\eta_1, \eta_2)$ of the corresponding Lagrangian satisfies the equation:

$$\mathbf{b}^*(\eta_1, \eta_2) = \hat{\mathbf{b}} - \lambda_1 \partial_{\mathbf{b}} \|\mathbf{b}^*\|_2 + (\eta_1 - \eta_2)D^{-1}\mathbb{1},$$

where $\partial_{\mathbf{b}} \|D\mathbf{b}^*\|_2$, the sub-gradient of $\sqrt{\sum_{p'=1}^{P}(\sum_{p=1}^{P}\mathbf{b}[p])^2}$ is given by:

$$\partial_{\mathbf{b}[p]} \|\mathbf{b}^*\|_2 = \begin{array}{ll} \frac{\mathbf{b}^*[p]}{\|\mathbf{b}^*\|_2} & \text{if} \quad \|\mathbf{b}^*\|_2 > 0 \\ (-1, +1) & \text{if} \quad \|\mathbf{b}^*\|_2 = 0 \end{array}$$

Therefore, given values of $\eta_1$ and $\eta_2$ and defining $\beta = (\eta_1 - \eta_2)D^{-1}\mathbb{1}$, we get

$$\mathbf{b}^*[p](\eta_1, \eta_2) = \begin{cases} \hat{\mathbf{b}}[p] + \beta[p] - \lambda\frac{\hat{\mathbf{b}}[p]+\beta[p]}{\|\hat{\mathbf{b}}+\beta\|_2}, & \text{if} \quad \left\|\hat{\mathbf{b}} + \beta\right\|_2 > \lambda \\ 0 & \text{otherwise} \end{cases}$$

which gives:

$$
\mathbf{a}^*(\eta_1, \eta_2) = \begin{cases} \hat{\mathbf{a}} + (\eta_1 - \eta_2)\mathbb{1} - \lambda \dfrac{\hat{\mathbf{a}} + (\eta_1 - \eta_2)\mathbb{1}}{\left\| \mathrm{D}^{-1}\hat{\mathbf{a}} + (\eta_1 - \eta_2)\mathbb{1} \right\|_2} & \text{if} & \left\| \mathrm{D}^{-1}\hat{\mathbf{a}} + (\eta_1 - \eta_2)\mathbb{1} \right\|_2 > \lambda \\ \qquad\qquad = 0 & \text{otherwise} \end{cases}
$$
(22)

And the gradient of eqn. (21) with respect the dual variables at the dual-feasible solution is :

$$
\nabla_{\eta_1}\mathcal{L} = -\mathbb{1}^\top \mathbf{a}
$$
$$
\text{and } \nabla_{\eta_2}\mathcal{L} = \mathbb{1}^\top \mathbf{a}
$$

Therefore, the dual-ascent step is

$$
\eta_1^{(k+1)} = \eta_1^{(k)} - \alpha^{(k)} \min\{\mathbb{1}^\top \mathbf{a}, 0\}
$$
$$
\eta_2^{(k+1)} = \eta_2^{(k)} + \alpha^{(k)} \max\{\mathbb{1}^\top \mathbf{a} - 1, 0\}
$$

Therefore using the fact that violation of constraints for $\eta_1$ and $\eta_2$ are mutually exclusive, we combine $\eta_1$, $\eta_2$ into a single dual variable $\eta \in \mathbb{R}$, giving the the dual feasible solution as:

$$
\mathbf{a}^{(n+1)} = \begin{cases} \hat{\mathbf{a}} + \eta^{(n)}\mathbb{1} - \lambda \dfrac{\hat{\mathbf{a}} + \eta^{(n)}\mathbb{1}}{\left\| \mathrm{D}^{-1}\hat{\mathbf{a}} + \eta^{(n)}\mathbb{1} \right\|_2} & \text{if} & \left\| \mathrm{D}^{-1}\hat{\mathbf{a}} + \eta^{(n)}\mathbb{1} \right\|_2 > \lambda \\ \qquad\qquad 0 & \text{otherwise} \end{cases}
$$
(23)

and the dual-ascent step is:

$$
\eta^{(n+1)} = \begin{cases} \eta^{(n)} - \alpha^{(n)}\mathbb{1}^\top \mathbf{a}^{(n)} & \text{if} & \mathbb{1}^\top \mathbf{a}^{(n)} < 0 \\ \eta^{(n)} + \alpha^{(n)}(\mathbb{1}^\top \mathbf{a}^{(n)} - 1) & \text{if} & \mathbb{1}^\top \mathbf{a}^{(n)} > 1 \end{cases}
$$
(24)

## F   Dependence on Initialization

In order to demonstrate the robustness of the solution to initialization, here we show the result of estimation procedure for different initializations of the $K$-means step. As mentioned earlier, the the initial positions of the $K$-means centroids are uniformly distributed at random. It can be observed that regardless of the $K$–means solution the algorithm converges to highly consistent global graphical structures. Note that for each solution, labels values have been selected to maximize overlap across results to improve comparison.

## G   Cross-Validation

A 10-fold block cross validation approach is used for model selection and to assess the performance of the various models. Here the time-series $\mathbf{y}[t]; 1 = 1 \ldots T$ is divided into 10 contiguous blocks of length $T/10$ each and the model parameters estimated using 9 blocks. The data from the remaining block (indexed as $t = 1 \ldots T'$) are the fitted to the model using least squares to get $\hat{\mathbf{y}}$. For example for the standard order-$P$ VAR model with parameters $\mathrm{A}[1] \ldots \mathrm{A}[P]$, this would be

$$
\min_{\hat{\mathbf{y}}} \sum_t \left\| \hat{\mathbf{y}}[t] - \mathbf{y}[t] \right\|_2
$$
$$
\text{such that} \quad \hat{\mathbf{y}}[t] = \sum_{p=1}^{P} \mathrm{A}[p]\hat{\mathbf{y}}[t-p]
$$

Defining $\eta[t]; t = 1 \ldots T$ as the Lagrange multiplier for each constraint and $\rho > 0$ as the augmented Lagrangian multiplier term, the solution for this is computed using a dual-ascent procedure, where

(a) K-Means initialization

(b) Estimated graphical structure

(c) K-Means initialization

(d) Estimated graphical structure

(e) K-Means initialization

(f) Estimated graphical structure

Figure 1: The leftcolumn shows the node assignments after one $K$–means initialization step. The right column shows the estimated graphical structure for the corresponding initialization. The nodes in the latent global graph are positioned at the centroids of the corresponding local graphs. In all figures, colors and labels both indicate cluster assignments of the sites. Labels for the local nodes are omitted for clarity.

each iteration involves the following two steps:

$$[\text{Primal update}]: \quad \hat{\mathbf{y}}[t]^{(k+1)} = \mathbf{y}[t] - \frac{1}{2}\left(\eta^{(k)}[t] - \sum_{p=1}^{P} A[p]^{\top}\eta^{(k)}[t+p]\right)$$

$$[\text{Dual update}]: \quad \eta[t]^{(k+1)} = \eta[t]^{(k)} + \rho\left(\hat{\mathbf{y}}^{(k+1)}[t] - \sum_{p=1}^{P} A[p]\hat{\mathbf{y}}^{(k+1)}[t-p]\right)$$

Estimating the best least-squares fit for the hierarchical model, given parameters $A, B$ and $Z$ and hyper-parameter $\lambda_0$, involves solving

$$\min_{\hat{\mathbf{y}},\mathbf{x}} \sum_t \|\hat{\mathbf{y}}[t] - \mathbf{y}[t]\|_2 + \left\|\mathbf{x}[t] - \lambda_0 \sum_{p=1}^{P} A[p]\mathbf{x}[t-p]\right\|_2$$

$$\text{such that} \quad \hat{\mathbf{y}}[t] = \sum_{q=1}^{Q} B[q]\hat{\mathbf{y}}[t-p] + Z\mathbf{x}[t]$$

which can be solved using the following dual-ascent scheme:

$$[\text{Primal update}]: \quad \hat{\mathbf{y}}[t]^{(k+1)} = \mathbf{y}[t] - \frac{1}{2}\left(\eta^{(k)}[t] - \sum_{q=1}^{Q} B[q]^{\top}\eta^{(k)}[t+p]\right)$$

$$[\text{Dual update}]: \quad \eta[t]^{(k+1)} = \eta[t]^{(k)} + \rho\left(\hat{\mathbf{y}}^{(k+1)}[t] - \sum_{q=1}^{Q} B[p]\hat{\mathbf{y}}^{(k+1)}[t-p] - Zx^{(k+1)}[t]\right)$$

while $\mathbf{x}^{(k+1)}$ is estimated using the method described in Supplemental Appendix D.

The relative error is then defined as

$$\sqrt{\frac{\sum_t \|y[t] - \hat{y}[t]\|}{\sum_t \|y[t]\|}}$$