[Reviews · NeurIPS 2014]

Submitted by Assigned_Reviewer_15

This paper proposes a model of spatio-temporal dynamics that models
a global latent process that governs the interactions between
high level clusters of points together with a local observed process
in which interactions are decoupled from points outside of one's cluster.
Both levels can be thought of as vector autoregressive models.
The authors apply their method to modeling data from a numerical
simulation of a geologic model of fluid flow under the earth's subsurface.

In general, this was a technically strong paper and shows off some state of the art
optimization techniques. On the other hand it was also a difficult paper to get through as
it was notation heavy and quite dense. In particular, there are topics
that are not typically ``NIPS'' topics, for which I think the authors should spend
more time providing intuition for.
As an example, I do not have a good grasp
what it means to approximate the latent
process with a 1-dimensional convection diffusion system --- the authors talk about
the computational benefits --- but what kind of behavior can we not capture
when we make this approximation?

The experimental setting seemed interesting but I would have liked to see some more
discussion about the task --- how was the training/test split performed in
this setting? And how are errors measured? (The authors report errors in percentage,
but it is unclear what the denominator here is).

I would have also like to see more extensive experiments, particularly with more than one dataset,
as it is difficult to understand if the proposed model is generally applicable or if it
is specifically good for this fluid-flow setting. Though spatio-temporal processes
that have interesting things happening at multiple scales are ubiquitous, the paper
does make assumptions (such as the 1-dimensional convection-diffusion assumption again)
that make the specific optimization approach tractable --- but the authors do not
discuss other settings in which these assumptions might also apply.

typos:
Next in Section 3, relaxations to simply the problem along with with efficient algorithm
for parameter estimation are developed.
unbalanced bracket on line 184

page 5 second to last line (268) --- parentheses at wrong level for alpha^{(n)}

grep for nonhierarchal and hierarchal -> nonhierarchical, hierarchical

Summary: This is a technically strong paper that proposes a multiscale spatiotemporal model of
flow through porous media. The optimization algorithm is quite interesting as well
as the results.

Submitted by Assigned_Reviewer_26

This paper introduces a new graphical model for multi-scale analysis of spatio-temporal dynamic systems.

The contribution of the paper relies on a new prior distribution for the parameters of the multi-scale process.

The paper is generally well written and provides an original and significant contribution to the literature. My major concern is about the exposition in Introduction and Section 3. Since the main contribution regards the derivation of the penalization term, I would have expected a discussion on the relationships with the following penalization methods: LASSO, elastic net, and SCAD, and an illustration of the advantages of the proposed method over the other methods.
Summary: I would expect the authors introduce a discussion on the advantages of their prior with respect to other penalization methods. I would strongly suggest to accept the paper.

Submitted by Assigned_Reviewer_39

This paper presents an approach to modeling latent state linear dynamical systems, with a focus on discovering hierarchical structure in the dynamics. To achieve this end, the authors propose an optimization approach where the link between the latent (global) and local dynamic dynamics are encouraged to have a certain group sparsity pattern. The authors show that this procedure is able recover the underlying connectivity structure of a fluid flow PDE.

Overall, I believe this to be a good paper, and one that that should be accepted to NIPS. However, there are a also a few points that I believe could strengthen the paper. The underlying model here seems to be fairly simple and one that most of the NIPS audience will understand, but the presentation here is tied heavily to the convection-diffusion equations, which are likely to be much less understandable to many in the audience. Ultimately, this connection seems to boil down to how we penalize the latent state dynamics (the "D(gamma) A" term in the regularization function g), but this connection was still a bit tenuous to me. Since this appears to be one of the key elements of the paper (the other penalty function were just the normal group LASSO or LASSO penalty, from relaxations of the sparse structure), I wish this particular element had been explained more fully. Given that this seems to be the chief algorithmic contribution of the paper over normal group-lasso time series modeling, the current paragraph surrounding eq (3) is a little bit dense, and I think will be hard to understand for the majority of the NIPS audience as well.

Given this, I think highlighting further one of the overall "big ideas" of the paper, that learned dynamical systems can perform substantially better using regularization derived from the underlying physical process, is very important to stress further, and could ultimately be the biggest take-home message of the paper for most of the NIPS audience.
Summary: The paper proposes a new optimization approach for developing hierarchical systems based upon convection-diffusion dynamics. The key elements here a little bit hard to parse, but I believe there is a good idea here, which is worth publishing.
Author Feedback
Author rebuttal: We would like to thank the reviewers for their positive reviews and insightful comments.

To address the concerns raised, we shall include more discussion on the interpretation, intuition and limitations of the convection-diffusion approximation. And, we shall emphasize the advantages of physically motivated priors in this problem. We shall also fix the typos in the revision.

A 10-fold block cross validation approach is used, where the time-series is divided into 10 contiguous blocks and parameters estimated using 9 blocks. The data y_t in the remaining block are fitted to the model using least squares to get \hat{y}_t. For VAR models, this involves projecting the data on to the VAR linear system, and for the hierarchical model also requires estimating x* as explained in Sec 3.1. The relative fitting error is then calculated as sum_t [ (hat{y}_t – y_t)/y_t ]. We will clarify this in the revision.

Regarding more experiments, due to space limitations and the need for clear exposition of the problem setting and methodology, we are constrained to report results for only one example. We have tested the model on a number of synthetic problems and the results are qualitatively similar. This example was chosen as it illustrates many of the features encountered in exploration geophysics (http://www.spe.org/web/csp/datasets/set02.htm). We are currently preparing a journal article with exhaustive validation on synthetic and real data from fluid flow and electromagnetic propagation studies. Going forward, we also intend to test our method on biological, neural and social networks.

The paper includes discussion and experimental evaluation wrt the standard group-LASSO method for time-series graph structure learning. To the best of our knowledge, LASSO, SCAD and EN have not been reported on this problem and it is unclear how to apply them in a meaningful fashion, that captures the spatial and temporal structure of the problem. We will include a discussion on alternative penalization strategies and their pros and cons.